# GOAL: A Generalist Combinatorial Optimization Agent Learner

**Darko Drakulić, Sofia Michel & Jean-Marc Andreoli**
NAVER LABS Europe, France
`firstname.lastname@naverlabs.com`

## Abstract

Machine Learning-based heuristics have recently shown impressive performance in solving a variety of hard combinatorial optimization problems (COPs). However they generally rely on a separate neural model, specialized and trained for each single problem. Any variation of a problem requires adjustment of its model and re-training from scratch. In this paper, we propose GOAL (for Generalist combinatorial Optimization Agent Learner), a generalist model capable of efficiently solving multiple COPs and which can be fine-tuned to solve new COPs. GOAL consists of a single backbone plus light-weight problem-specific adapters for input and output processing. The backbone is based on a new form of mixed-attention blocks which allows to handle problems defined on graphs with arbitrary combinations of node, edge and instance-level features. Additionally, problems which involve heterogeneous types of nodes or edges are handled through a novel multi-type transformer architecture, where the attention blocks are duplicated to attend the meaningful combinations of types while relying on the same shared parameters. We train GOAL on a set of routing, scheduling and classic graph problems and show that it is only slightly inferior to the specialized baselines while being the first multi-task model that solves a wide range of COPs. Finally we showcase the strong transfer learning capacity of GOAL by fine-tuning it on several new problems. Our code is available at this url.

## 1 Introduction

Combinatorial Optimization (CO) problems are extensively studied in the Operations Research community, because they are both challenging in terms of theoretical complexity and central in many applications such as transportation, logistics and finance. Traditional CO methods include exact solvers, typically based on integer programming (Korte & Vygen, 2018), and approximation algorithms (Vazirani, 2003) which provide optimality or approximation guarantees, respectively, but do not scale; and heuristic algorithms (Boussaïd et al., 2013) which produce good-quality solutions faster but without guarantees. Effective heuristics heavily rely on expert knowledge and are generally problem-specific. A more recent and promising family of methods for CO is based on Machine Learning, either to learn the hyper-parameters of classic CO solvers, or to directly learn heuristics in an end-to-end fashion (Bengio et al., 2021; Cappart et al., 2021). In particular constructive neural heuristics have been successful at solving a variety of CO problems, including routing (Bello et al., 2017; Nazari et al., 2018; Kool et al., 2018; Kwon et al., 2020), scheduling (Zhang et al., 2020; Kwon et al., 2021) and graph problems (Khalil et al., 2017; Qiu et al., 2022). However, in most cases, a separate model is specialized and trained for each problem, and sometimes even for each distribution of instances within a problem, especially for varying instance sizes. While much effort has been devoted to improving the generalization to different instance distributions (Manchanda et al., 2022; Zhou et al., 2023; Son et al., 2023; Drakulic et al., 2023), generalization to different problems is less studied. Indeed, although there are strong neural heuristics for many CO problems, any variation of these problems requires adjustment of their model and re-training from scratch. While it is true that some problem-specific model tailoring may be required, since the structure of problem instances and solutions are very diverse, many problems are related and it would be interesting to exploit their commonalities. In this paper, we seek to answer the following question: *Can we design and train a single generalist model so that it can be efficiently adapted to solve a*

*variety of CO problems?* The idea of pretraining a unified model on some pretext task so that it can be efficiently fine-tuned to several downstream tasks has been extremely successful in the language and vision domains (Devlin et al., 2019; Brown et al., 2020; Radford et al., 2021). In these domains, a self-supervised pretraining task exploits the intrinsic structure and regularities of images and text to learn generic representations. In contrast, in CO, while most problem instances can be modeled as graphs, there is no generic regularity to leverage by a pretext task. Graphs, unlike images or text contain little redundancy or intrinsic semantics.

In this paper, we propose a new model, named GOAL (for Generalist combinatorial Optimization Agent Learner), which consists of a single backbone and light-weight problem-specific adapters for input and output processing. To handle the large discrepancy between CO problems, we propose three complementary strategies. First, to ensure learning related representations for different CO problems, we propose to use a problem-specific projection of the input instance features to a small fixed-size representation, which is then projected into the main embedding space using a *shared codebook*. Second, to handle problems defined on graphs with arbitrary combinations of node, edge and instance-level features, we design a new form of *mixed-attention blocks* for the backbone, injecting edge information at the core of each attention block. Finally, we introduce a novel *multi-type transformer* architecture to deal with problems which involve heterogeneous types of nodes or edges (typically represented as multipartite graphs). The idea is to duplicate the backbone's mixed-attention blocks in order to attend each meaningful combination of types while relying on the same fixed set of shared parameters. In principle, GOAL is able to handle any CO problem whose instances can be represented as a graph and where feasible solutions can be constructed iteratively, as a sequence of node selections. We find that this pattern is generic enough to cover a wide range of problems. We train GOAL in a multi-task setting, by imitation of expert trajectories of eight classic CO problems, spanning multiple domains: *Routing problems*: the Asymmetric Traveling Salesman Problem, the Capacitated Vehicle Routing Problem with or without Time Windows, the Orienteering Problem ; *Scheduling problems*: the Job Shop Scheduling Problem and the Unrelated Machine Scheduling Problem; *Packing problems*: the Knapsack Problem; and *Graph problems*: the Minimum Vertex Covering Problem. The expert trajectories are based on good-quality solutions, easily generated by traditional problem-specific solvers for instances of medium size (generally 100 nodes).

Experimentally we first show that GOAL performs very well on all the training tasks, making it the first model that solves such a variety of CO problems. Second, we consider eight new problems and demonstrate that GOAL can be fine-tuned to reach a good performance, either in a few minutes using few-shot imitation learning or in a couple of hours without supervision. In both cases, the fine-tuning takes a fraction of the time and data needed to train a model from scratch. We further provide an ablation study to evaluate the effectiveness of a shared codebook, multi-type transformer architecture and mixed-attention blocks. Notably the GOAL architecture gives state-of-the-art results on 7 out of our 8 training tasks when used as single-task model. In summary, our contributions are the following:

- We introduce GOAL: a generic and flexible model for end-to-end combinatorial optimization. Two key ingredients of GOAL are the simple yet effective form of *mixed-attention blocks* which enables handling problems represented by graphs with any combination of node-, edge- and instance-level features and its *multi-type transformer* architecture which allows to seamlessly tackle problems with multipartite graph structures.
- We demonstrate that it is possible to learn a single backbone model to effectively solve a variety of CO tasks, spanning routing, scheduling, packing and graph problems.
- We showcase the potential of such a multi-task pretrained model to be efficiently adapted to handle varied new CO tasks, with or without supervision. We believe this is a key step toward a foundation model for CO.

## 2 RELATED WORKS

**End-to-end heuristic learning for CO** Our work falls into the category of end-to-end constructive heuristics for combinatorial optimization. Constructive neural heuristics have been successfully applied to a wide range of COPs, including many variants of routing problems (Kool et al., 2018; Xin et al., 2021; Kim et al., 2022; Kool et al., 2022; Fu et al., 2021; Sun & Yang, 2023), scheduling

problems (Zhang et al., 2020; Kwon et al., 2021), as well as classic COPs such as the Knapsack Problem (Kwon et al., 2020) and the Maximum Independent Set (Khalil et al., 2017; Sun & Yang, 2023). Another successful category of neural CO methods are improvement heuristics, which focus on iteratively enhancing a given complete solution, typically by using a neural network to guide the selection of local search operators (Chen & Tian, 2019; Hottung & Tierney, 2020; Ma et al., 2021; Wu et al., 2022). The seminal work of Khalil et al. (2017) introduces graph embeddings which do work for a variety of CO problems but with limited performance. Current state-of-the-art models are tailored for certain problem structures (e.g. euclidian routing problems, where nodes are represented by their coordinates plus some extra node or graph features such as demand or vehicle capacity), and successfully handled by transformer-based architectures (Kool et al., 2018). For non-euclidian routing problems, Kwon et al. (2021) and Drakulic et al. (2023) propose adapted architectures to deal with the distance matrix, but they are specialized for such problems and do not apply to those with node-only features. In GOAL, we propose a novel mixed-attention modules which can flexibly deal with both node and edge features, incurring no overhead when one or the other is missing.

**Generalization of neural heuristics**  While the first seminal models for neural CO had a dramatic drop in performance when tested on out-of-training distribution instances (Joshi et al., 2022; Manchanda et al., 2022), several recent approaches show a greatly improved generalization performance, especially to larger instances (Fu et al., 2021; Li et al., 2021; Son et al., 2023; Pan et al., 2023), to distribution shifts of other instance parameters (Bi et al., 2022), or both (Manchanda et al., 2022; Bi et al., 2022; Zhou et al., 2023; Drakulic et al., 2023). Concurrently with our work, some papers have started to explore cross-problem generalization for the family of routing problems. Specifically, (Lin et al., 2024) propose to pretrain the AM model (Kool et al., 2018) on the TSP and to fine-tune either the entire model or some light-weight modules on the Price Collecting TSP and OP. Additionally, Zhou et al. (2024); Liu et al. (2024); Berto et al. (2024) explore generalization to different variants of the CVRP. They build on the idea of zero-shot compositionality (Ruis et al., 2021) by training a model on CVRPs with some combinations of constraints from a predefined set (e.g. time windows and duration limit constraints) and show that the model generalizes to some unseen combinations of the same constraints. Wang & Yu (2023) propose a bandit based approach to alternate between problems during training, so as to focus on those which have the best return. Although these approaches tackle multiple problems, it is important to note that the considered Euclidian routing problems remain very similar in structure and indeed many of those problems were solved by the seminal Attention Model of (Kool et al., 2018) with little adaptations of its base model. Our paper goes further and considers multi-task training and transfer learning of a new model on a variety of COPs with different structures.

**Multi-task training for sequential decision making**  Multi-task (pre-)training has recently shown promising results in related fields. For example, Ibarz et al. (2022) proposed a graph neural network processor capable of learning to execute a variety of algorithms, such as sorting and searching. The paper focuses only on polynomial problems and learns to imitate the *execution* of classical algorithms. In contrast, we address NP-hard problems and learn by imitation of the *solution* provided by (potentially exponential) specialized solvers. Recent approaches in control rely on sequence modeling of expert trajectories from multiple tasks (Stooke et al., 2021; Sun et al., 2023). Although they treat various tasks, with potentially diverse state and action spaces, these works still consider image observations and exploit this common structure. Reed et al. (2022) propose GATO, a general-purpose agent obtained by training a large transformer-based model on hundreds of tasks spanning vision, language and control. It demonstrates the potential of a single network (albeit with 1.2B parameters) to effectively tackle a wide variety of tasks.

## 3  MULTI-TASK LEARNING FOR COMBINATORIAL OPTIMIZATION

### 3.1  CONSTRUCTIVE CO PROBLEM SOLVING

We assume that a CO problem instance is given by a pair of a finite, non-empty set $X$ of *feasible solutions*, and a real-valued *objective* function $f$ whose domain contains $X$. Solving the instance consists in finding an *optimal solution*, i.e. an element of $\arg\min_{x \in X} f(x)$. Constructive approaches assume that $f$ is defined not only over $X$ but over a set of *partial solutions*, which includes $X$ and is amenable to a step-wise construction procedure where each partial solution can be

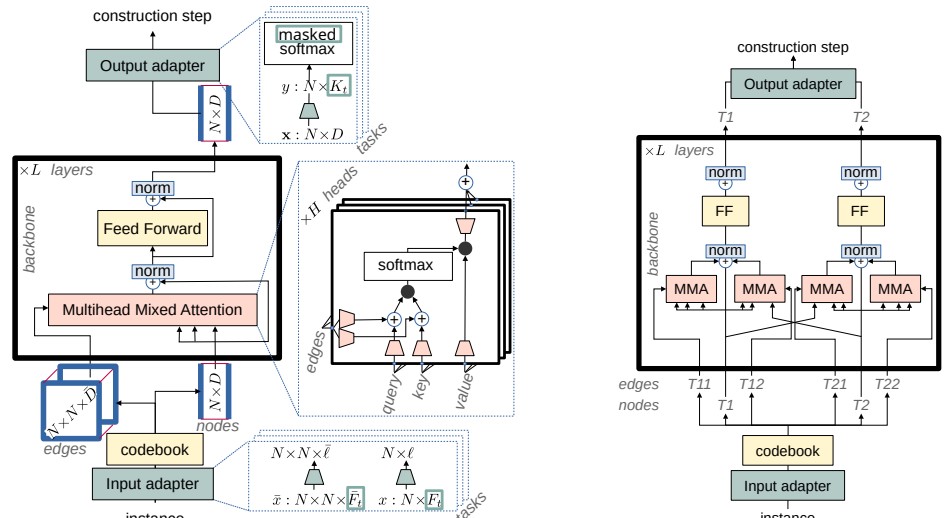

Figure 1: Left: Architecture of GOAL for single-type problems; the green components are task-specific, while the others belong to the backbone. The codebook is a shared $\ell \times D$ (resp. $\bar{\ell} \times \bar{D}$) matrix (with $\ell \ll D, \bar{\ell} \ll \bar{D}$) applied to the node (resp. edge) representations produced by the input adapters. Right: Architecture of GOAL for multi-type problems (with here 2 node types T1,T2); it uses multiple mixed attention blocks ("MMA"), some in self- and some in cross-attention mode, but they all share the **same** parameters (per layer); so do the two Feed Forward blocks ("FF"). For the edges, all the type combinations are shown (here T11,T12,T21,T22) but those which have no meaning in a task are simply omitted. The same set of parameters is used for the two architectures.

obtained by a sequence of construction steps. Thus, solving the problem instance can be formulated as a sequential decision procedure which iteratively selects the construction steps so as to reach not only a feasible but an optimal solution. That in turn can be modeled as a Markov Decision Problem (MDP) associated with the problem. In this paper, we use the BQ-MDPs introduced by Drakulic et al. (2023) which are defined for "tail-recursive" problems, i.e. the widespread class of problems that are solvable by dynamic programming (Bertsekas, 2019). In BQ-MDPs, the actions are the construction steps, typically the selection of a node, while the states are simply instances. This differs from many peer NCO models where each state is a pair of an instance and a partial solution. The tail recursion property precisely ensures that the two alternatives lead to "bisimilar" MDPs, which have the same optimal policies, as shown in Drakulic et al. (2023), but the BQ-MDPs' states more efficiently capture the relevant information to determine these policies.

## 3.2 MULTI-TASK CO PROBLEMS

Now, given a class $\mathcal{T}$ of tail-recursive CO problems called tasks, let $\boldsymbol{\Omega}_t$ for each task $t \in \mathcal{T}$ denote the parametric space used to represent its instances. Consider the new (multi-task) CO problem whose instances are represented in the disjoint union $\biguplus_{t \in \mathcal{T}} \boldsymbol{\Omega}_t$ consisting of pairs $(t, \omega)$ of a task identifier $t \in \mathcal{T}$ and a point $\omega \in \boldsymbol{\Omega}_t$ in its corresponding parameter space. It is easy to show that the multi-task problem is tail-recursive whenever each of its components is. And its BQ-MDP is then simply the disjoint union of the BQ-MDPs of its components.

In this paper, we study alternatives for the multi-task *policy* space, beyond the (trivial) Cartesian product of the policy spaces of the individual tasks, where no parameter is shared between the different tasks. The underlying assumption is that if the tasks are sufficiently *similar*, learning one task may elicit implicit skills useful to other tasks, which are then leveraged by parameter sharing. In the extreme, some tasks may require little training data and/or just a few shot fine-tuning to obtain acceptable performance, leveraging their similarity to more thoroughly trained tasks. Since each task has its own parametric space used to represent its instances, some part of any multi-task policy has to be task specific, and trained by processing at least a few instances of each task: pure zero-shot generalization is not possible. The situation is close to multi-lingual Natural Language Processing,

where the vocabulary of each language needs to be seen at least once at training or fine-tuning. And we borrow the solution used in that context, namely adapters (Bapna & Firat, 2019).

### 3.3 TASK REPRESENTATION

To leverage operational similarity between tasks, we need them to have at least representational similarity. Thus, we assume that each instance of size $N$ of a task (CO problem) is represented by a set of $N$ abstract "nodes", together with features which can be attached at node, edge (node pair) or instance level. Instance features can be replicated and appended to the features of each node, hence can be ignored without loss of generality. Since the construction steps (actions) are returned by the task-specific output adapters (Sec 3.4), there are no restrictions on their format. For simplicity and to cover the problems used in our experiments, we assume in the sequel that each construction step consists of the selection of a node, possibly with an option among a fixed number $K$ of options (dependent on the task: in most cases, $K{=}1$ and the action simply selects a node).

Hence, the GOAL model takes three inputs $t, x, \bar{x}$, where $t$ is a task identifier, and $(x, \bar{x}){\in}\mathbf{\Omega}_t$ is a parameter representing an instance of task $t$, where $x{\in}\mathbb{R}^{N \times F_t}$ (resp. $\bar{x}{\in}\mathbb{R}^{N \times N \times \bar{F}_t}$) represents the feature vectors of dimension $F_t$ of the $N$ nodes (resp. $\bar{F}_t$ of the $N{\times}N$ edges). While some tasks may have no edge feature ($\bar{F}_t{=}0$), we always assume that at least one node feature exists, which distinguishes the nodes by a randomly assigned value between 0 and 1, similar to the random features used in Graph Neural Networks (Sato et al., 2021), so we always have $F_t{\geq}1$. Both $F_t$ and $\bar{F}_t$, as well as the meaning of the corresponding features, depend on task $t$. The output of the model is a matrix $y{\in}\mathbb{R}^{N \times K_t}$ representing the score of each candidate action, i.e. the pair of a node and an option for selection. As usual, it is passed to a softmax (over the whole matrix) to obtain a probability distribution. Wherever appropriate, task-dependent masking can be used to ensure that probabilities are null for the actions disallowed by a task.

### 3.4 ARCHITECTURE OF THE MODEL

The architecture of our model generalizes that of BQ-NCO (Drakulic et al., 2023), and is described in Fig. 1 left. It essentially consists of a sequence of layers whose parameters are shared by all the tasks (backbone) together with task-specific adapter modules at the input and output of the model[1]. The input-output adapter modules for task $t$ are specific to the dimensions $F_t, \bar{F}_t, K_t$ and the meaning of the corresponding features. No positional encoding is used: all relational information is assumed to be captured by $\bar{x}$. The layers themselves, on the other hand, are only aware of the shared embedding dimensions $D, \bar{D}$. They are self-attention transformer layers (Vaswani et al., 2017) where vanilla attention is replaced by multi-head *mixed* attention, similar in intent to (Kwon et al., 2021; Henderson et al., 2023) but with a different design.

Our mixed attention block, in the general case where $M$ nodes attend on $N$ nodes ($M{=}N$ only in self-attention), is an instance of generalized attention (Andreoli, 2019). A generalized attention mechanism takes the following inputs: node representations as vectors for $N$ queries, $M$ keys, and $M$ values, with an optional $M{\times}N$ log-binary mask $\mathcal{M}$. In each head $h$, each query, key, value vector is first projected into a common space of dimension $d$, yielding for each $n{=}1{:}N$ and $m{=}1{:}M$, vectors (of dimension $d$):

$$ Q_n^{(h)} = Q_n \mathbf{W}_Q^{(h)}, \qquad K_m^{(h)} = K_m \mathbf{W}_K^{(h)}, \qquad V_m^{(h)} = V_m \mathbf{W}_V^{(h)}. $$

Then the key and query vectors are used to compute an $M{\times}N$ score matrix $S$, which is transformed into an attention matrix (by adding the mask and applying softmax) multiplied by the reprojected value vectors to produce the output representation $r$ of the $N$ input query nodes as a matrix with $N$ rows:

$$ r = \sum_h \underbrace{\mathrm{softmax_{col}}(S^{(h)}{+}\mathcal{M})^\top}_{\text{attention matrix}} V^{(h)} \mathbf{W}_O^{(h)\top}. $$

Attention mechanisms then differ in the way they compute the score matrix:

$$ \text{(vanilla attention)} \quad S_{mn}^{(h)} = \langle\, K_m^{(h)} \mid Q_n^{(h)} \,\rangle, \tag{1a} $$

$$ \text{(our mixed attention)} \quad S_{mn}^{(h)} = \langle\, K_m^{(h)}{+}K_{mn}'^{(h)} \mid Q_n^{(h)}{+}Q_{mn}'^{(h)} \,\rangle. \tag{1b} $$

---

[1] Adapters can also be introduced within layers, but this paper considers only input-output adapters.

In vanilla attention, the scores are obtained by simple scalar product of the key and query vectors (equation 1a). Our mixed attention mechanism takes as additional input $M \times N$ "edge" vectors, projected twice into the common space of dimension $d$, once as key and once as query, resulting in vectors (of dimension $d$):

$$Q'^{(h)}_{mn} = E_{mn}\mathbf{W}'^{(h)}_Q, \qquad\qquad K'^{(h)}_{mn} = E_{mn}\mathbf{W}'^{(h)}_K. \qquad (2)$$

The score matrix is still obtained as a scalar product of a key and a query vector, but now, each of them is the sum of a node component and an edge component (equation 1b). Our mixed attention allows to incorporate arbitrary relational information (edge features) into the core of the attention mechanism. It differs from positional encoding, which attempts to instill a priori the edge information (limited to sequential ordering) into the node representations.

In the GOAL backbone, all the node (resp. edge) inputs to the attention blocks are of dimension $D$ (resp. $\bar{D}$), and they are all projected into a common space of dimension $d$ equal to $D$ divided by the number of heads. No masking is used in the backbone. One well-known limitation of this model is its quadratic complexity, the attention matrices being of size $N^2$. In practice though, for many tasks, a simple, task-specific heuristic can deal with that problem by pre-filtering the set of nodes to present to the model (usually much fewer than $N$). Whenever available, we use such heuristics.

Input and output adapters are linear projections, from the task-specific instance features to the backbone's input embeddings and from the backbone's output embeddings to the task-specific action space. However, when the embedding dimension is large and the number of training tasks is small, there is a risk that the embeddings produced by each task end up occupying complementary subspaces of the embedding space throughout the layers, which would defeat the purpose of effectively sharing parameters across tasks. To prevent this, we simply force the input linear projection to be low rank. Thus, node (resp. edge) features are first mapped into a low dimension $\ell \ll D$ (resp. $\bar{\ell} \ll \bar{D}$), through a task-specific linear projection (the input adapter proper), forcing the representations to share dimensions across tasks. Then this "small" representation is plunged into a "large" $D$- (resp. $\bar{D}$)-dimensional embedding, through a common linear projection, called a *codebook*, shared by all the tasks (see Fig. 1 left).

### 3.5 Multi-Type Transformer Architecture

Many tasks (CO problems) involve different types of nodes. For example, scheduling problems often involve "operation" and "machine" nodes. For GOAL, one solution is to simply treat them all indiscriminately in the backbone. The adapters, being problem specific, are of course free to treat them differently according to their types. This is sometimes unsatisfactory though, since it forces the adapter to provide uniform embeddings for all the nodes and edges, though they are often heterogeneous by nature: in the scheduling example, edge features may carry precedence constraints when between operations, or processing times when between machines and operations, and no information at all when between machines. Instead of packing all these different types of relations into a single vector, with padding values wherever a feature is undefined for a given edge, we propose a *multi-type architecture* where the adapters output separate embeddings per type (for nodes) or compatible pairs of types (for edges). Each layer of the model, instead of being a monolithic self-attention block operating on all the nodes, becomes a task-specific combination of both cross- and self-attention blocks, as described by equation 1b, operating on different type-compatible subsets of nodes and edges. This is illustrated in Fig. 1 right in the case of two types. For $n$ types, the architecture would be similar but involves $n^2$ mixed-attention blocks, where each type attends to itself and every other type, or a selection of relevant types defined by the task. Therefore, in our scheduling example, the model will contain separate self-attention blocks, for operation nodes and machine nodes respectively, and cross-attention blocks, for operation-machine edges and vice-versa. Observe that while the architecture configuration is now task-specific, the model parameters remain the same for all the configurations: all the attention blocks of a layer, be them self- or cross-attentions, and whatever the types they involve, share the same, layer-specific parameters. This is precisely what allows us to train a unique model to solve arbitrary single- and multi-type problems.

### 3.6 Training Procedure

We train GOAL on a set of eight various CO problems (described in Sec 4). For each training problem, we define the corresponding BQ-MDP, the linear input and output adapters to match the

dimensions of the input features and the construction actions. We train the model by imitation of high-quality solutions, provided by specialized solvers. At each step, we sample a problem and create a batch of instances by sampling suffix subproblems from the training instances, similarly to (Drakulic et al., 2023). The forward and backward processes involve the input and output adapters, as well the backbone parameters, where the mixed-attention and feed-forward blocks are duplicated within each layer if the sampled task is multi-typed.

## 4 EXPERIMENTS

Our experiments aim to address the following questions: (i) does the GOAL approach enable learning a single backbone to effectively solve a variety of CO problems (Sec 4.1)? (ii) can it be used as a pretrained model and efficiently fine-tuned to solve new CO problems, unseen at training (Sec 4.2)? (iii) to what extent do our mixed-attention blocks and multi-type architecture contribute to the results (Sec 4.3)? We begin by describing the multi-task training of GOAL.

**Training tasks.** We train GOAL on eight classic and varied CO problems. For each problem, we generate a training dataset of 1 million random instances and solve them using traditional solvers. Specifically, we consider the following problems and *oracle* solvers: the Asymmetric Traveling Salesman Problem (ATSP) with LKH (Helsgaun, 2017), the Capacitated Vehicle Routing Problem (CVRP) with LKH (Helsgaun, 2017), the Capacitated Vehicle Routing Problem with Time Windows (CVRPTW) with HGS (Vidal, 2022) the Orienteering Problem (OP) with A4OP (Kobeaga et al., 2018), the Knapsack Problem (KP) with ORTools (Perron & Furnon, 2022), the Minimum Vertex Covering Problem (MVC) with FastWVC (Cai, 2023), the Unrelated Machine Scheduling Problem (UMSP) with HiGHS (Huangfu & Hall, 2018) and the Job Shop Scheduling Problem (JSSP) with ORTools (Perron & Furnon, 2022). The training instances are of size 100 nodes, or $10 \times 10$ for JSSP and $100 \times 20$ for UMSP (resulting in both cases in 100 operation nodes). The description of the problems and the associated data generation processes is provided in Appendix F.

**Training details.** GOAL is trained by alternating batches in random order from each of the training tasks, using a single-class cross-entropy loss for ATSP, CVRP, CVRPTW, OP, JSSP and UMSP and a multi-class cross-entropy loss for KP and MVC. We use AdamW optimizer (Loshchilov & Hutter, 2018), with an initial learning rate of 0.0005 and a decay rate of 0.97 every 10th epoch. We trained the model on 8 Nvidia V100 GPU servers, with batch size of 256 for 7 days, completing ca. 400 epochs. The best model was selected based on the average performance on a validation set of 128 instances per problem. The model hyperparameters are described in Appendix A.

**Baselines.** We consider the following constructive neural models as baselines: AM (Kool et al., 2018), MDAM (Xin et al., 2021), POMO (Kwon et al., 2020), Sym-NCO (Kim et al., 2022), BQ-NCO (Drakulic et al., 2023), MVMoE (Zhou et al., 2024), MatNet (Kwon et al., 2021), S2V-DQN (Khalil et al., 2017), RouteFinder (Berto et al., 2024), COMPASS (Chalumeau et al., 2023) and Gumbeldore (Pirnay & Grimm, 2024). All models except MVMoE and RouteFinder are specialized and trained in single-task mode. They are often combined at test time with various enhancers, such as beam search Drakulic et al. (2023), simulation-guided beam search (Choo et al., 2022), MCTS (Xing & Tu, 2020), instance augmentation (Kwon et al., 2020), or active search (Hottung et al., 2021), but for fair comparison, all the results we report are obtained by a pure greedy application of the trained policy at test time.

### 4.1 PERFORMANCE ON THE TRAINING TASKS

In Table 1 we compare the performance of GOAL to the relevant neural baselines for each problem as well as its single-task version (i.e. GOAL trained separately for each task). First, we observe that the best performance is achieved by the single-task GOAL in 7 out of the 8 tasks, with a computation time comparable to that of the baselines. This proves the versatility and effectiveness of our proposed architecture, even for a specialized model. Second, we see that multi-task GOAL is only slightly worse than the single-task version. These results validate the capability of the GOAL backbone, together with the task-specific adapters, to effectively solve the variety of training tasks. More broadly, they demonstrate that multi-task learning can be a successful paradigm for neural combinatorial optimization.

| | ATSP100 | | CVRP100 | | CVRPTW100 | | OP100 | |
|---|---|---|---|---|---|---|---|---|
| | gap | time | gap | time | gap | time | gap | time |
| Oracle solver | 0.00% | 29s | 0.00% | 12m | 0.00% | 10m | 0.00% | 1.1m |
| AM greedy | - | | 7.11% | 1s | | | 5.35% | 1s |
| MDAM greedy | - | | 4.84% | 1s | - | | 2.88% | 16s |
| POMO no aug | - | | **1.21**% | 1s | - | | - | |
| Sym-NCO greedy | - | | 3.33% | 1s | - | | 2.03% | 2s |
| BQ-NCO greedy | 1.27% | 2s | 2.79% | 2s | - | | 0.22% | 2s |
| MVMoE/4E | - | | 1.65% | 1s | 4.90% | 1s | - | |
| RouteFinder-TE | - | | 1.50% | 1s | 3.19% | 1s | - | |
| MatNet greedy | 0.93% | 1s | - | | - | | - | |
| **GOAL** SINGLE-TASK **greedy** | **0.30**% | 10s | 2.34% | 10s | **2.61**% | 10s | **-0.04**% | 3s |
| **GOAL** MULTI-TASK **greedy** | 0.91% | 10s | 3.16% | 10s | 3.82% | 10s | 0.43% | 3s |
| | KP100 | | MVC100 | | UMSP100x20 | | JSSP10x10 | |
| | gap | time | gap | time | gap | time | gap | time |
| Oracle solver | 0.00% | 1s | 0.00% | 2m | 0.00% | 2m | 0.00% | 47s |
| POMO no aug | 0.19% | | - | | - | | - | |
| BQ-NCO greedy | **0.10**% | | - | | - | | - | |
| S2V-DQN | - | | 0.97% | 2s | - | | - | |
| COMPASS | - | | - | | - | | 4.70% | 3h |
| Gumbeldore greedy | - | | - | | - | | 3.17% | 9s |
| **GOAL** SINGLE-TASK **greedy** | **0.10**% | 3s | **0.23**% | 3s | **2.82**% | 7s | **2.73**% | 15s |
| **GOAL** MULTI-TASK **greedy** | 0.12% | 3s | 0.37% | 3s | 3.84% | 7s | 4.13% | 15s |

Table 1: Performance on 128 test instances from the training tasks. Each gap is with respect to the oracle and often corresponds to the optimality gap. Lower is better.

In addition, in Appendix E, we provide experimental results demonstrating GOAL's excellent zero-shot generalization on instances up to 10 times larger than the training instances. We can note in particular the strong generalization for ATSP which seems much more challenging for the state-of-the-art baselines. This confirms the effectiveness of our mixed-attention mechanism.

## 4.2 FINE-TUNING TO NEW TASKS

In this section, we want to evaluate the transfer capability of the trained GOAL backbone to unseen tasks. We consider eight diverse new COPs: the Traveling Repairman Problem (TRP), the Prize Collecting TSP (PCTSP), the Open CVRP (OCVRP), the Split Delivery CVRP (SDCVRP), the Sequential Ordering Problem (SOP), the Maximum Covering Location Problem (MCLP), the Maximum Independent Set (MIS) problem and the Open Shop Scheduling Problem (OSSP) (detailed descriptions in Appendix F). GOAL can be fine-tuned to specific problems in two modes. In both, the adapters are trained from scratch, and the backbone is also open to backpropagation (no parameter freeze). The supervised mode is identical to training, except it processes only a few (labeled) instances of the problem for a few steps. Details and results are given in Appendix B. The unsupervised mode is more challenging, but also more realistic, since we may not have access to good-quality solutions for the new problem. We propose to fine-tune the model by iterative imitation-improvement steps between an Apprentice and an Expert, as in ExIt (Anthony et al., 2017). While Exit relies on a possibly complex tree-search for improvement, we use instead a brute-force but fast method, where the Expert samples multiple solutions (in our case 128) per instance from the Apprentice and selects the best, if it is sufficiently better, and sends it back to the Apprentice (current model) for imitation. This is similar to the self-labeling strategy proposed by Corsini et al. (2024) for the JSSP. One could also use more sophisticated experts such as generic local search heuristics or smarter sampling (Pirnay & Grimm, 2024) to accelerate the finetuning. Performance results are given in Fig 2, which shows that unsupervised fine-tuning systematically outperforms unsupervised training of the whole model from scratch at a comparable processing time budget. We also compare finetuning GOAL to finetuning the single-task models and report the results in Fig 4. We observe that the multi-task model consistently performs as well or better than the "most-related" single-task

model (e.g. CVRP model for OCVRP or JSSP model for OSSP), hinting to the synergistic effect of the multi-task training.

Figure 2: Unsupervised fine-tuning of GOAL to eight diverse new tasks. We report the best results over 10 runs both for training from scratch and fine-tuning.

## 4.3 ABLATION STUDY

**Mixed-attention blocks.** We compare our mixed-attention blocks with alternative modules which also aim at mixing the edge information with the attention scores, namely MatNet (Kwon et al., 2021) and G2G (Henderson et al., 2023). More precisely we focus on the ATSP and train three versions of GOAL with the three different variants of mixed-attention. For MatNet, we keep the original hyper-parameters used for solving the ATSP. Fig 3 (left) presents the models performance (optimality gap on a test set of 128 instances) at different points of the training. We see that using our mixed-attention blocks leads to a significant gain, on the performance at convergence (MatNet) or on the speed of convergence (G2G). Note that these ablations are performed on a smaller training dataset than in the previous experiments (100k instances), hence the larger optimality gap.

**Multi-type transformer.** Our multi-type architecture is meant to deal with tasks that are naturally represented by multipartite graphs with heterogeneous nodes or edges, typically scheduling problems where job and machine nodes play different roles. To evaluate its effect on the performance, we compare it to that obtained with the more direct alternative which ignores node and edge heterogeneity: we simply gather in a single tensor the node (edge) representations of the different types, produced by the input adapter, and use the original single-type model. Note that because of the linearity of the input adapters, these representations can be obtained either by defining a specific linear transform per type, or equivalently by defining a single linear transform on the direct sum of the feature spaces of the different types. We implemented the former for the Job Shop Scheduling Problem and present in Fig 3 (middle) a comparison between the training performance of the single-type and multi-type architectures. Clearly, the multi-type architecture converges faster and to a better performance: after 200 epochs of training on 100K instances, it reaches a test optimality gap of 4.6% while the single-type model achieves only 8.9%. Finally, note that the multi-type model offers more flexibility for fine-tuning: while, in order to be task agnostic at training, parameter sharing between the type specific attention blocks of a layer must be enforced, that constraint can be relaxed at fine-tuning to better fit the task at hand.

**Sparsification using a codebook.** Fig 3 (right) experimentally confirms the importance, in the fine-tuning phase (supervised mode), of sparsifying the input adapter with a shared codebook. The results are given for the PCTSP, but similar results are available in Fig 6 (Appendix) for the other problems. The shared codebook results in a more stable finetuning (over 10 runs) and to a better optimality gap. While we do not observe a significant effect on the performance on the training tasks, we believe that constraining the rank of the codebook tends to prevent its vectors from specializing for individual tasks seen at training, which would be of limited use when fine-tuning to new tasks.

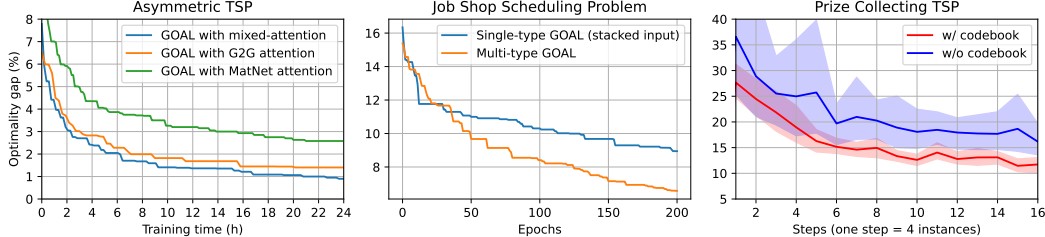

Figure 3: (left) Performance with different variants of mixed-attention. (middle) Performance of single- vs multi-type GOAL. (right) Fine-tuning of GOAL with and without the codebook.

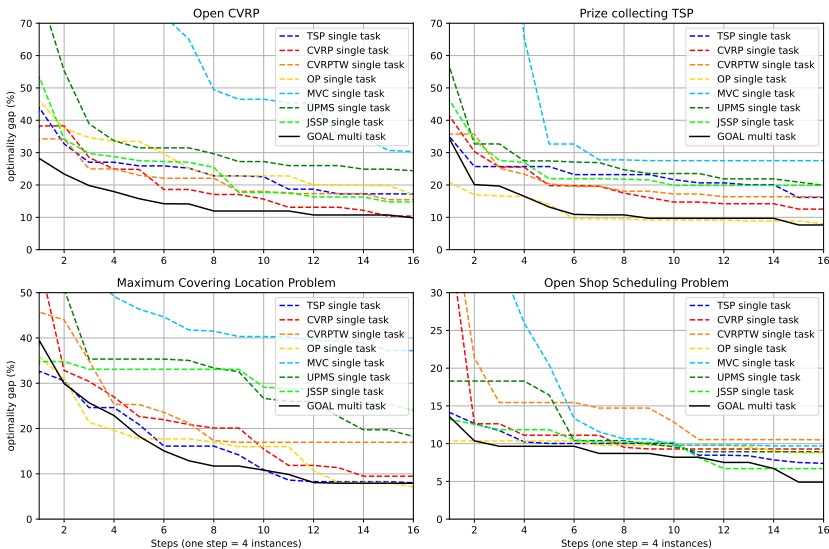

Figure 4: Comparison of the (supervised) fine-tuning of the single- versus multi-task GOAL on four new problems.

## 5 DISCUSSION AND CONCLUSION

In this paper, we propose a generalist neural CO model (GOAL) and demonstrate the feasibility of learning a shared backbone model to solve a variety CO problems, spanning routing, scheduling, packing and classic graph problems. We have shown the benefits of GOAL as (i) a single-task specialized model, providing state-of-the-art results on seven CO problems; (ii) a multi-task model, able to solve different problems with the same backbone; and (iii) a pretrained model, which is efficiently fine-tuned to new problems. The proposed multi-task pre-training relies on imitation learning and therefore the availability of good-quality solutions. While this is not an intrinsic limitation of the GOAL model, we believe that imitation learning is an effective training paradigm for learning multi-task models. This is indeed demonstrated by previous works which relied on sequence-modeling training tasks for control (Sun et al., 2023; Reed et al., 2022). It is particularly advantageous for CO as we can leverage high-quality solutions to classical NP-hard problems, provided by traditional solvers, which are based on decades of OR research. In contrast, for a new potentially more original problem, we show that GOAL can be effectively finetuned without any supervision. Our approach is limited to problems where solution feasibility is easy to ensure during the construction, which is a common requirement of constructive heuristics. Finally the quadratic complexity of the underlying transformer architecture makes the approach too costly for problem sizes beyond a thousand nodes. Exploring more efficient forms of attention as basis for our mixed-attention blocks would be a promising direction for better scaling.

ETHICS STATEMENT

To the best of our knowledge, no aspect of our research raises ethics issues. It did not involve human subjects, and considers only generic Combinatorial Optimization problems with no dubious association.

REPRODUCIBILITY STATEMENT

For full reproducibility, the implementation of the backbone, the adapters and all the experiments described in this paper are publicly available at `https://github.com/naver/goal-co`. The training data generation process, the training process, and its hyper-parameters are described in detail in the Appendix.

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

## A  GOAL HYPERPARAMETERS AND SIZE

The GOAL backbone consists of $L=9$ layers with an embedding size of $D=\bar{D}=128$ (the same for both nodes and edges), a feed-forward dimension of 512, and ReZero normalization Bachlechner et al. (2021). The mixed-attention blocks have 8 heads. The task specific input adapters produce representations of low dimension $\ell=8$ for nodes and $\bar{\ell}=4$ for edges, which are mapped to embeddings by a shared codebook of dimension $\ell \times D$ for nodes and $\bar{\ell} \times \bar{D}$ for edges, respectively. GOAL backbone has 2.1M parameters and adapters are very light, with only a few thousand parameters per task. For the illustration, RouteFinder (Berto et al., 2024) with POMO has 1.3M, MVMoE (Zhou et al., 2024) has 3.7M (the same model is used in RouteFinder with MVMoE), and RouteFinder with the Transformer Encoder 1.7M.

## B  SUPERVISED FINE-TUNING OF GOAL

In section 4.2 we describe the process of fine-tuning our multitask model on eight unseen tasks in unsupervised mode. Although fine-tuning without labeled data is useful for transferring to new tasks, it comes at a cost - it is quite expensive. Our experiments show that fine-tuning to the new problems with unsupervised learning takes, on average, a few hours.

On the other hand, we demonstrate that GOAL can be effectively fine-tuned using a very small amount of labeled data, in a short time. Figure 5 presents the performance of fine-tuning GOAL on the same eight problems used in unsupervised tuning. Here, we use only 128 solved (but not necessarily optimal) instances and tune the model with just one training step per instance (more precisely, one training step per tail-subproblem of each instance). In contrast to unsupervised tuning, this method is extremely fast; fine-tuning takes only a few minutes.

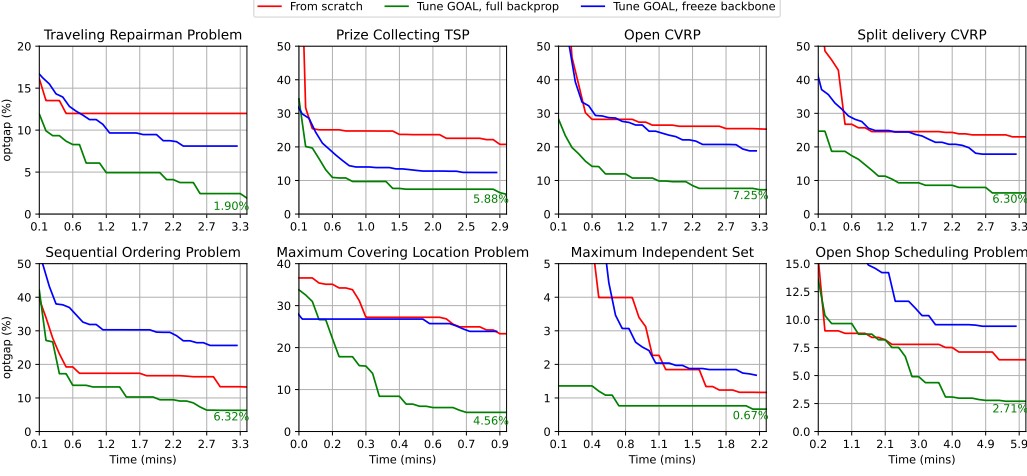

Figure 5: Supervised fine-tuning on eight new problems. We report the best results over 10 runs for both training from scratch and fine-tuning.

## C  IMPACT OF CODEBOOK ON FINE-TUNING

In 4.3 We discussed the importance of sparsifying the input adapter with a shared codebook. Although this feature does not impact training performance, it is very important in the fine-tuning phase. By adding a codebook to the model, fine-tuning becomes more stable and achieves better performance. In the main text, we presented a figure comparing fine-tuning for the PCTSP with models that have and do not have a codebook. Figure 6 illustrates the impact of the codebook on fine-tuning the multitask model across all eight problems we used for transfer. In these experiments, we used light-trained models with and without a codebook. The models are trained on 100K instances, for 200 epochs. For each experiment, we perform 10 inference runs and report the mean with a confidence interval of 100.

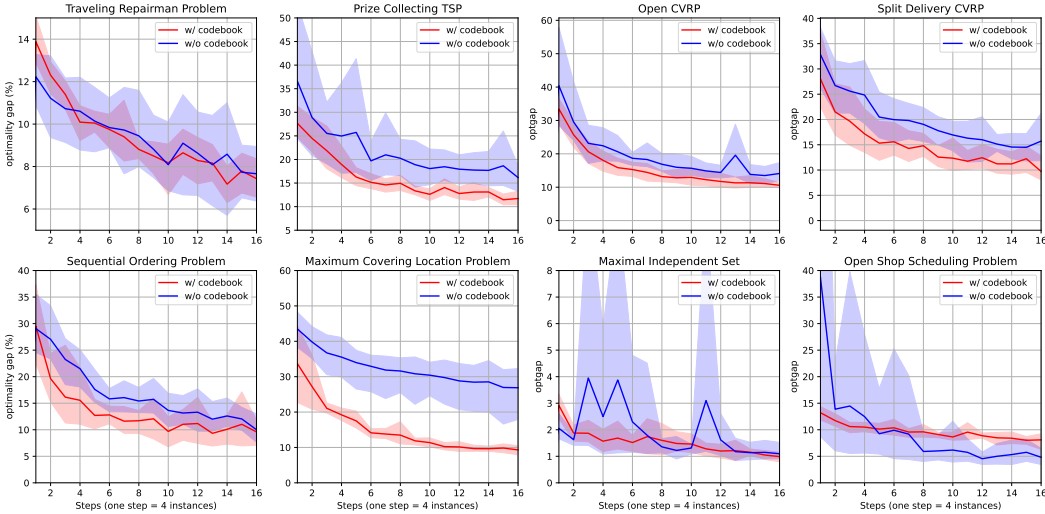

Figure 6: Ablation study of codebook

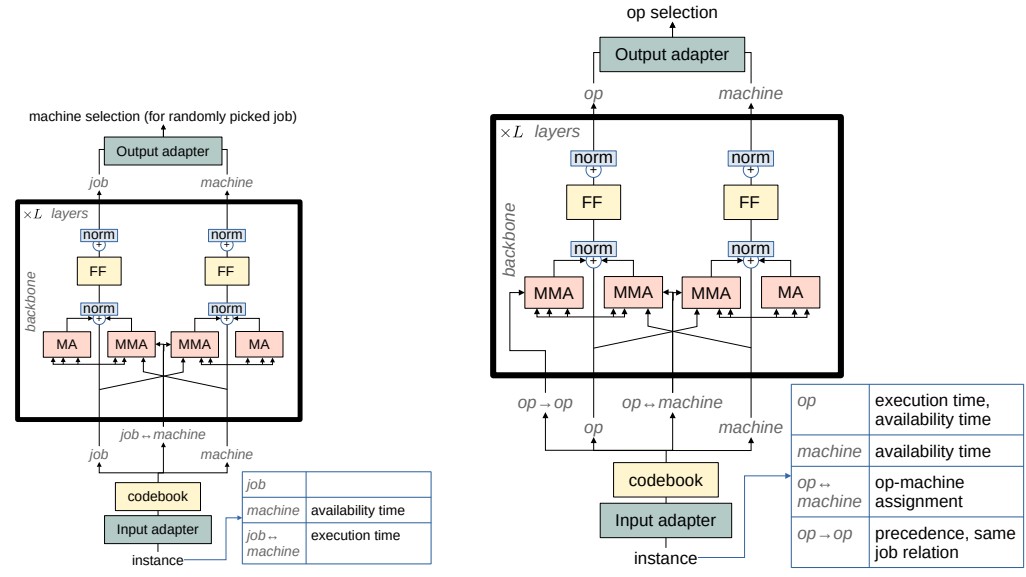

(a) the Unrelated Machine Scheduling Problem (UMSP)

(b) the Job-shop Scheduling Problem (JSSP)

Figure 7: Illustration of the multi-type model on scheduling tasks in GOAL. The "MA" blocks use vanilla multi-head attention while the "MMA" use our multi-head mixed-attention.

# D    MULTI-TYPE MIXED ATTENTION LAYERS FOR UMSP AND JSSP

As an illustration of our proposed multi-type transformer architecture, we present in Figure 7 the inputs to the mixed-attention blocks for the Unrelated Machine Scheduling (UMSP) and the Job Shop Scheduling Problems (JSSP). Note that in the case of UMSP, since there is no edge information between jobs and between machines, the self-attention blocks (left-most and right-most attention blocks) degenerate to vanilla attention. The cross-attention blocks, on the other hand, are mixed, and they are fed the same edge information, holding the job execution times on each machine (transposed from each other). In JSSP, the situation is similar, but ops have two kinds of relations between themselves: precedence, and sharing the same job. Hence the self-attention block on ops is mixed.

| | | Train. distrib. 100 | | Generalization 200 | | 500 | | 1000 | |
|---|---|---|---|---|---|---|---|---|---|
| ATSP | LKH | 0.00% | 29s | 0.00% | 48s | 0.00% | 2m | 0.00% | 4m |
| | MatNet | 1.57% | 1s | 97.21% | 12s | - | | - | |
| | BQ-NCO | 1.63% | 1s | 1.90% | 9s | 4.29% | 26s | 8.09% | 1m |
| | **GOAL** | 0.45% | 5s | 1.54% | 30s | 1.62% | 3m | 1.96% | 5m |
| CVRP | LKH | 0.00% | 12m | 0.00% | 2h | 0.00% | 5h | 0.00% | 7h |
| | AM | 7.11% | 1s | 10.16% | 1s | 17.2% | 1s | 72.95% | 2s |
| | MDAM | 4.84% | 1s | 7.54% | 6s | 12.94% | 24s | 32.25% | 1m |
| | POMO | 1.21% | 1s | 5.94% | 6s | 31.36% | 32s | 284.58% | 1m |
| | Sym-NCO | 3.33% | 1s | 8.42% | 3s | 37.58% | 54s | 481.53% | 9m |
| | BQ-NCO | 2.79% | 1s | 2.81% | 9s | 3.64% | 55s | 5.88% | 2m |
| | **GOAL** | 3.16% | 5s | 3.42% | 30s | 3.90% | 2m | 7.37% | 5m |
| CVRPTW | HGS (1s) † | *108/128* | 2m | *14/128* | 2m | *0/128* | 2m | *0/128* | 2m |
| | HGS (1m) | 0.00% | 10m | 0.00% | 26m | 0.00% | 2h | 0.00% | 2h |
| | **GOAL** | 3.82% | 5s | 7.71% | 30s | 9.49% | 2m | 6.75% | 5m |
| OP | EA4OP | 0.00% | 1m | 0.00% | 3m | 0.00% | 11m | 0.00% | 45m |
| | AM | 5.35% | 1s | 11.05% | 1s | 22.29% | 1s | 30.73% | 1s |
| | MDAM | 2.88% | 16s | 5.55% | 1s | 18.31% | 4s | 30.05% | 15s |
| | SymNCO | 2.03% | 2s | 6.60% | 3s | 19.81% | 24s | 31.79% | 1m |
| | **GOAL** | 0.43% | 3s | 1.60% | 10s | 7.45% | 17s | 15.62% | 30s |
| KP | ORTools | 0.00% | < 1s | 0.00% | < 1s | 0.00% | < 1s | 0.00% | < 1s |
| | POMO | 0.19% | 1s | 0.50% | 1s | 6.41% | 31s | 5.34% | 1m |
| | BQ-NCO | 0.10% | 2s | 0.14% | 6s | 0.74% | 7s | 0.92% | 13s |
| | **GOAL** | 0.12% | 3s | 1.63% | 6s | 2.40% | 10s | 2.59% | 16s |
| | | 10x10 | | 10x15 | | 15x15 | | 20x15 | |
| JSSP | ORTools | 0.00% | 47s | 0.00% | 2m | 0.00% | 3h | 0.00% | 80h |
| | COMPASS ◇ | 4.70% | 3h | - | | 8.00% | 5h | 10.40% | 8h |
| | **GOAL** | 4.13% | 15s | 11.02% | 43s | 10.74% | 45s | 22.53% | 4m |

Table 2: Generalization performance on larger instances. Lower is better. All NCO methods perform greedy decoding, except for methods marked with ◇; they do not provide results for greedy decoding. Solvers marked with † are unable to solve all instances within the given time budget; we report the number of instances successfully solved.

# E    GENERALIZATION TO NEW INSTANCE DISTRIBUTIONS

Table 2 presents the generalization to instances up to 10 times bigger than the training ones, for a subset of the training tasks. Our approach takes a bit more time than some baselines, but note that scaling to much larger instances is not a focus of this paper. One possible way to avoid the quadratic complexity of attention is by using task-specific heuristics, such as the KNN heuristic for routing problems (proposed by Drakulic et al. (2023), to reduce the size of the input instance and the overall complexity. Another approach could be using methods that do not rely on quadratic attention blocks, such as those proposed by Luo et al. (2024), though this method tends to be highly specialized for specific problems.

Our model demonstrates excellent generalization performance in terms of the optimality gap for problems with various edge features, such as ATSP (with an asymmetric distance matrix) and JSSP (with two binary matrices with precedence constraints and job-machine relations).

# F    DESCRIPTION OF THE CONSIDERED CO PROBLEMS

In this Section, we describe the CO problems (in alphabetical order) that are solved by GOAL as well as the instance generation process. GOAL solves problems from various categories: Routing, Scheduling, Packing, Graph, and Location Problems. For each problem, we provide all the details of how it is represented in our model, including a list of instance, node and edge features, the action chosen by the model, and the objective. We also describe how we generate a solution trajectory for problems where we use single cross-entropy loss for behavioral cloning.

### F.1 CAPACITATED VEHICLE ROUTING PROBLEM AND SPLIT-DELIVERY CAPACITATED VEHICLE ROUTING PROBLEM

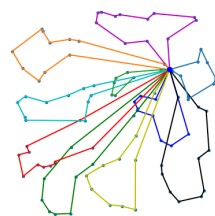

The Capacitated Vehicle Routing Problem (CVRP) is a standard routing problem. The goal of CVRP is to optimize the delivery routes of a fleet of vehicles to a set of customers while considering each vehicle's capacity. The objective is to minimize the overall distance traveled by the vehicles while respecting the capacity constraints and ensuring that all the customer demands are met. Split-Delivery CVRP (SDCVRP) is a variant of the CVRP that allows a single customer's demand to be fulfilled by multiple vehicles. In our experiments we use Euclidean CVRP for training GOAL and SDCVRP for fine-tuning.

**Data generation.** Node locations are randomly sampled from the unit square, while node demands are randomly generated integers from the interval [1, 10]. Following Kool et al. (2018), the vehicle capacity depends on the problem size: capacity of 50 for a problem size of 100, 80 for 200, 100 for 500, and capacity of 250 for problems with 1000 nodes.

| | |
|---|---|
| **Instance features**: | Remaining vehicle capacity |
| **Node features**: | Origin/destination tokens, demand |
| **Edge features**: | Distance between nodes |
| **Oracle**: | LKH |
| **Objective**: | Minimize traveled distance |
| **Action**: | Select a node, with direct edge or via-depot |
| **Trajectory**: | Nodes ordered by a solution subtour. Subtours are ordered by their final remaining capacity in descending order |

### F.2 CAPACITATED VEHICLE ROUTING PROBLEM WITH TIME WINDOWS

The Capacitated Vehicle Routing Problem with Time Windows (CVRPTW) is an extension of the CVRP. In addition to demand constraints, each customer also has a specified time window during which they can be visited, along with the necessary service time required for serving that customer. The objective of the CVRPTW is to find routes for the vehicles that satisfy the capacity limits, deliver goods within the specified time windows, and minimize the total travel and waiting time or the number of vehicles used. In our experiments, we consider the standard version of the Euclidean problem where total traveling and serving time should be minimized and we use it for training GOAL.

**Data generation.** For the time windows and service times, we use a similar procedure to Falkner & Schmidt-Thieme (2020), where the locations are sampled sampled from the unit square. We ensure the feasibility of the resulting instances by sampling time windows using a more refined procedure. Windows start time values are sampled from the range (0.01, 0.25). It's important to note that start times do not directly impact problem feasibility, as vehicles can wait at the node until the service starts. On the other hand, the choice of the visiting end time has an impact on feasibility. An instance is unfeasible if the window end time of a customer is less than the travel time from the depot plus the service time. To avoid this, end times are sampled from the interval [max(window start time, travel time from depot) + 2 × service time, max(window start time, travel time from depot) + 4 × service time].

| | |
|---|---|
| **Instance features**: | Remaining vehicle capacity, current time |
| **Node features**: | Origin/destination tokens, demand, start time window, end time window, service time |
| **Edge features**: | Distance between nodes |
| **Oracle**: | HGS |
| **Objective**: | Minimize traveled time |

| **Action**: | Select a node, with direct edge or via-depot |
| **Trajectory**: | Nodes ordered by a solution subtour. Subtours are ordered by their final remaining capacity in descending order |

### F.3 JOB SHOP SCHEDULING PROBLEM

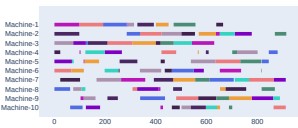

The Job Shop Scheduling Problem (JSSP) is a classic scheduling problem. We are given $N$ jobs and $M$ machines. Each job consists of a sequence of operations that need to be processed in a specific order (implying precedence constraints). Each operation has a given processing time. Each machine can only process one operation at a time and when an operation is started it cannot be interrupted (no preemption). A solution is a schedule, defined by the start time of each operation such that a specific criterion, such as makespan, total completion time, or total tardiness is minimized.

**Data generation.** We follow the standard procedure for generating JSSP benchmark datasets: execution times are randomly sampled from $U(1, 100)$.

| **Instance features**: | *None* |
| **Node features**: | Operation execution time, job availability time, machine availability time |
| **Edge features**: | Precedence constraint, operation-machine dependency |
| **Oracle**: | ORTools |
| **Objective**: | Minimize makespan |
| **Action**: | Select an operation |
| **Trajectory**: | Tasks ordered by minimal finishing time |

### F.4 KNAPSACK PROBLEM

The Knapsack Problem (KP) is a well-known combinatorial optimization problem that involves selecting a subset of items from a given set. Each item had a weight and a value, and the goal is to maximize the cumulated value of selected items, without exceeding a given cumulated weight threshold, known as the knapsack's (weight) capacity.

**Data generation.** Following the same procedure as in Kwon et al. (2020), node weights and values are randomly sampled from the unit interval, and knapsack capacity is set to 25.

| **Instance features**: | Remaining knapsack capacity |
| **Node features**: | Value, weight |
| **Edge features**: | *None* |
| **Oracle**: | ORTools |
| **Objective**: | Maximizing packed values |
| **Action**: | Select an item |
| **Trajectory**: | There is no order |

### F.5 MAXIMUM INDEPENDENT SET PROBLEM

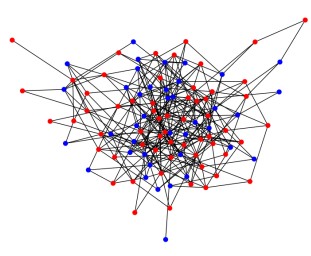

In the Maximum Independent Set (MIS), we are given an undirected graph $G = (V, E)$ and the goal is to find a subset of nodes $S \subset V$ of maximal size $|S|$ such that each pair of nodes in $S$ is not connected by an edge. In other words, the maximum independent set is the largest possible independent set in $G$, i.e., an independent set with the maximum number of vertices. In our experiments, we fine-tune GOAL to MIS problem.

**Data generation.** For this problem, we follow a similar procedure as in Khalil et al. (2017). We generate random

graphs by using Erdős–Rényi model with the critical edge probability $p$ sampled from [0.05, 0.15].

**Instance features**: *None*
**Node features**:     *None*
**Edge features**:     Adjacency value (0-1)
**Oracle**:            FastWVC
**Objective**:         Maximize number of selected nodes
**Action**:            Select an node
**Trajectory**:        There is no order

### F.6 MAXIMUM COVERAGE LOCATION PROBLEM

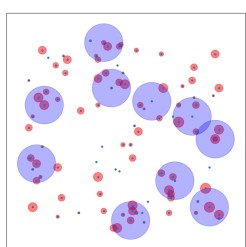

In the Maximum Coverage Location Problem (MCLP), we are given a set of locations with associated weights and several facilities with associated radius of coverage. The goal is to find the optimal placement of the facilities, on a subset of the locations, to maximize the number of covered locations. There is also a variant in which nodes have associated weights; in this case, the objective is to maximize the sum of the weights of the covered locations. In our experiments, we fine-tune our multitask model to MCLP.

**Data generation.** Node locations are randomly sampled from the unit square, the number of facilities is fixed to 10 and their radius of coverage to 0.1.

**Instance features**: Number of facilities, radius of coverage
**Node features**:     *None*
**Edge features**:     Distance between nodes
**Oracle**:            PuLP[2]
**Objective**:         Maximize the number of covered locations
**Action**:            Select an facility node
**Trajectory**:        There is no order

### F.7 MINIMUM VERTEX COVER PROBLEM

The goal of the Minimum Vertex Cover Problem (MVC) is to find the smallest set of vertices in a graph, such that each edge is incident to at least one of these vertices. In the weighted version of this problem (MWVC), each vertex has an associated weight, and the goal is to select vertices with the minimum cumulated sum of weights while satisfying the vertex cover condition. Data generation and problem features are the same as in MIS. We use MVC to train our model.

### F.8 OPEN SHOP SCHEDULING PROBLEM

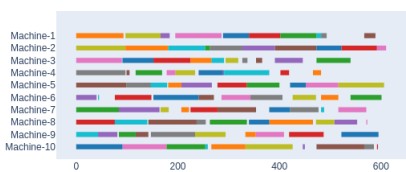

The Open Shop Scheduling Problem (OSSP) is a scheduling problem in which a set of jobs (with a given set of operations) must be processed on a set of machines, but unlike job shop scheduling, there is no predefined order. Each operation must be processed, with no overlapping between operations and machines at the same time unit. The goal is typically to minimize the makespan, or the total time required to complete all jobs. Data generation and problem features are the same as in JSSP, except OSSP does not have precedence constraints.

### F.9 OPEN VEHICLE ROUTING PROBLEM

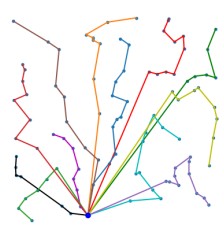

Open Vehicle Routing Problem (OCVRP) is a variant of CVRP where an unbounded fleet of vehicles is available at the depot, each serving a set of customers. In contrast with other CVRP variants, vehicles do not need to return to the depot at the end of their service. This makes OCVRP similar to the classical CVRP, as both problems involve a set of locations with demand and vehicle capacity constraints. However, the key difference lies in the route structure: in OCVRP, vehicles do not need to return to the depot after completing deliveries, which affects the solution space and cost minimization strategy. OCVRP has common applications in logistics, where delivery trucks do not need to return to the depot after completing deliveries. Data generation and problem features are the same as for the CVRP. In our experiments, we handle the Euclidean version of the OCVRP and fine-tune GOAL to solve it.

### F.10 ORIENTEERING PROBLEM

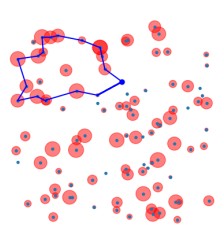

The Orienteering Problem (OP) is a combinatorial optimization problem that combines aspects of the Traveling Salesman Problem (TSP) and the Knapsack Problem. It is commonly used in routing and scheduling applications where the goal is to maximize collected rewards while respecting a constraint on total travel distance or time. The OP is defined by a set of locations (nodes), each associated with a reward; a starting location and an ending location (usually the same node); a travel cost matrix between locations; and a maximum allowable travel budget. The objective is to visit a subset of locations that maximizes the total collected reward while ensuring the total travel cost does not exceed the budget.

**Data generation.** We generated data following previous work Kool et al. (2018): node locations are uniformly sampled from the unit square and prizes are proportional to the distance to the depot (in Kool et al. (2018) this variant of the problem is called $OP_{distance}$). The maximal distance constraint is set to 4. We use Euclidean OP for training GOAL.

**Instance features**: Remaining distance
**Node features**:       Origin/destination tokens, prize
**Edge features**:       Distance between nodes
**Oracle**:              A4OP
**Objective**:           Maximize collected prizes
**Action**:              Select a node (including coming back to the depot)
**Trajectory**:          Ordered nodes by a solution tour

### F.11 PRICE COLLECTION TRAVELING SALESMAN PROBLEM

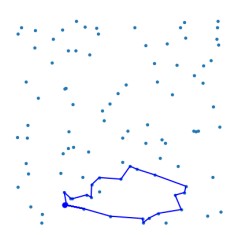

The Prize-Collecting Traveling Salesman Problem (PCTSP) is a variant of the classic Traveling Salesman Problem. In the PCTSP, each node has an associated prize and penalty, and the goal is to minimize the traveled distance while satisfying a minimum total prize constraint. Unlike the traditional TSP, where all cities must be visited, the PCTSP allows for skipping some nodes; however, unvisited nodes incur a penalty, creating a trade-off between collecting prizes and minimizing travel distance. The final objective is to minimize the

sum of the traveled distance and the sum of the penalties for unvisited nodes.

**Data generation.** We generated data following previous work Xin et al. (2021): node locations are uniformly sampled from the unit square, while prizes are randomly sampled from $U[0, num-nodes]$ and penalties are sampled from $U[0, num-nodes]$. Both features are normalized by multiplying by $4/num-nodes$ and $12/num-nodes$, respectively.

**Instance features**: Remaining prize constraint
**Node features**:   Origin/destination tokens, prize, penalty
**Edge features**:   Distance between nodes
**Oracle**:      ORTools
**Objective**:    Minimize the sum of traveled distances and penalties of univisited nodes
**Action**:      Select a node (including coming back to the depot)
**Trajectory**:    Ordered nodes by a solution tour

### F.12 SEQUENTIAL ORDERING PROBLEM

The Sequential Ordering Problem (SOP) is a combinatorial optimization problem where a set of tasks or nodes must be visited in a specific order while minimizing the total travel cost or distance. This problem can be modeled as the Traveling Salesman Problem, with precedence constraints. The objective is to find the shortest possible route that satisfies these precedence conditions. In our experiments, we fine-tune our multitask model to SOP.

**Data generation.** The values of the cost matrix are integers randomly generated from a uniform distribution $U[1, 1000]$ with zero costs between the same task. To fulfill feasibility constraints, we randomly order the tasks, and for the $i$-th task, we randomly add $k$ precedent jobs, where $k$ is sampled from $U[0, \lfloor i/2 \rfloor]$.

**Instance features**: *None*
**Node features**:   Origin token
**Edge features**:   Distance between nodes, precedence constraint
**Oracle**:      LKH
**Objective**:    Minimize cost
**Action**:      Select a node
**Trajectory**:    Ordered nodes by a solution tour

### F.13 TRAVELING REPAIRMAN PROBLEM

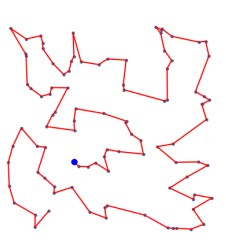

The Traveling Repairman Problem (TRP) is a combinatorial optimization problem where a repairman must determine the most efficient route to visit a set of locations, performing repairs or services at each site. In the Traveling Repairman Problem, a set of locations is given with the time needed to travel between any pair of locations. Each location has a task to be performed by the repairman. The objective of the problem is to find the route that minimizes the sum of the delays for reaching each point. In our experiments, we fine-tune GOAL to Euclidean TRP.

**Data generation.** Node locations are uniformly sampled from the unit square.

**Instance features**: *None*
**Node features**:   Origin token
**Edge features**:   Distance between nodes
**Oracle**:      LKH

**Objective**:      Minimize waiting time of all nodes
**Action**:      Select a node
**Trajectory**:      Ordered nodes by a solution tour

### F.14   TRAVELING SALESMAN PROBLEM

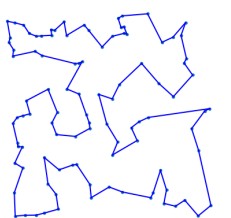

The Traveling Salesman Problem (TSP) is a classic combinatorial optimization problem. The objective of the TSP is to find the shortest possible route that a salesman can take to visit a given set of cities exactly once and return to its starting point. In the path-TSP variant, the origin and destination of the salesman are distinct nodes (but may have the same coordinates, so that the TSP is a special case of the path-TSP). A TSP instance can be represented either by node coordinates or by a distance matrix. The majority of NCO models address TSP in the coordinate setting, yet the matrix setting holds greater significance for practical applications, as it applies beyond the Euclidean context, where the matrix can represent time, energy, or cost, rather than distance, and need not satisfy the symmetry property of a distance matrix. In our work, we use the Asymmetric TSP (where the distances satisfy the triangular inequality property) to train our multi-task model."

**Data generation**   For the Euclidean case, we generate datasets as described in Kool et al. (2018): points are uniformly sampled from the unit square, and the corresponding distance matrix $A$ is computed. For non-Euclidean problems (ATSP) we generate data as described in Kwon et al. (2021), where distance matrices are generated randomly, but post-processed to satisfy the triangular inequality.

**Instance features**: *None*
**Node features**:      Origin/destination tokens
**Edge features**:      Distance between nodes
**Oracle**:      Concorde (for Euclidean) and LKH (for non-Euclidean)
**Objective**:      Minimize tour length
**Action**:      Select a node
**Trajectory**:      Ordered nodes by a solution tour

### F.15   UNRELATED MACHINE SCHEDULING PROBLEM

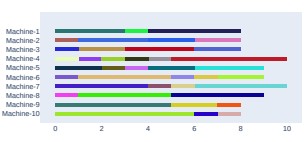

The Unrelated Machine Scheduling Problem (UMSP) is an optimization problem in operations research where a set of jobs must be assigned to a set of machines, each with different processing times for each job, to minimize a specific criterion, such as makespan, total completion time, or total tardiness. Unlike related machine scheduling, where machines have similar characteristics and processing times are proportional, in UMSP, the performance and efficiency of machines vary, making it a more complex and challenging problem to solve. Horowitz & Sahni (1976) shows that this problem is NP-hard. We use UMSP for multitask training of our GOAL model.

**Data generation.**   For UMSP, the processing times are randomly generated from $U(1, 100)$.

**Instance features**: *None*
**Node features**:      Machine availability times
**Edge features**:      Execution times
**Oracle**:      ORTools
**Objective**:      Minimize makespan
**Action**:      Select a machine to randomly selected task
**Trajectory**:      Ordered tasks by execution time

