# OpenReview forum: "GOAL: A Generalist Combinatorial Optimization Agent Learner"
_ICLR.cc/2025/Conference — ICLR 2025 Poster_

### Official Review · Reviewer_vQUg · 2024-10-28

**Soundness:** 3
**Presentation:** 3
**Contribution:** 2
**Rating:** 5
**Confidence:** 5

**Summary:**

The paper introduces GOAL, a generalist machine learning model for solving various combinatorial optimization problems. It uses a single backbone model with lightweight problem-specific adapters and shows strong performance in solving multiple optimization tasks. The model also exhibits efficient transfer learning capabilities.

**Strengths:**

Writing is easy to follow. Motivation makes sense. Experiments are sufficient.

The paper gives an interesting trial on the multi-task neural CO solver with a common encoder and task-specific adaptor.

**Weaknesses:**

## Contributions

- Contributions can be over claimed. Authors claim that it is ``the first model that solves such a variety of CO problems.'' As I know, [1] proposes a multi-task solver where encoders for different tasks share a common part, just like GOAL. So GOAL might not be the first one. Many other works e.g. [2] also have a similar idea. So, the idea of a multi-task solver may not be that new.

[1] Efficient training of multi-task combinarotial neural solver with multi-armed bandits. arXiv preprint arXiv:2305.06361, 2023.

[2] Multi-Task Learning for Routing Problem with Cross-Problem Zero-Shot Generalization

- The contribution to the foundational neural model for CO appears to be somewhat limited, as it seems to utilize the same neural network architecture (mixed attention) as MatNet.

## Experiments

- The running efficiency is not competent compared with other methods, as shown by Table 1 and 2. In ATSP and CVPR, it may take several more times of the running time compared with other methods (AM, MatNet, MDAM, etc.)

- There is still a significant gap between GOAL and the optimal solutions as shown in Table 1 and 2. Though it is understandable that some heuristics can be simultaneously fast and effective, the gap between GOAL and the best solver may make GOAL useless in solving real-world problems. Such a gap may also suggest that the time of ``general solvers'' has not yet arrived, and more efforts should be paid on task-specific models.

- Since GAOL adopts a similar neural networks as MatNet, the experimental results may not be that convincing. MatNet is trained on i.i.d. ATSP data, while GOAL is trained on multi-problem data. Given the same (or close) model capacity, how and why does it happen that MatNet performs worse than GOAL? Is it because that MatNet is trained with less epochs and does not converge completely? It looks quite unusual and requires a more detailed analysis and explanation.

## Theoretical analysis

- The neural network of mixed attention has a fatal flaw. The dimension of the node feature $x$ is $ N \times F_t$. When there are no explicit node features, $x$ is initialized as a one-hot vector. That is to say, in this case, the scale of the problem $N$ that the model can handle cannot exceed $F_t$, otherwise there would be a reuse of one-hot vectors. It appears that the model's generalization is severely compromised when $ N > F_t$, and the authors have not analyzed this situation.

- Though the method empirically works, there is still a lack of necessary theoretical explanations that why the method has the ability to generalize to instances of a larger scale while other baselines do not have such an ability. Also, there lacks comprehensive analysis over the method's multi-problem solving ability and how the model balances the weights during the learning process for different problems simultaneously to make the model performs best on average.

**Questions:**

- The mixed attention works similar to MatNet. So why MatNet cannot be run on ATSP500 or larger, but GOAL can? Another question, why GOAL outperforms MatNet, which parts of GOAL contribute to the improvement?

---

> ### Author Response · Authors · 2024-11-25
> **Rebuttal to original review**
>
> We appreciate the reviewer for their time and feedback on our work. We address your concerns as follows:
>
> > Contributions can be over claimed. Authors claim that it is ``the first model that solves such a variety of CO problems.'' ...  [1] and [2] already propose a multi-task solver ...
>
> We agree that the idea of multi-task training is not new, what we claim as new is the ability of our model to solve a wide variety of CO problems. Indeed the mentioned references tackle a limited set of problems (3 euclidian routing problems and the KP for [1] (now discussed in Sec 2) and several variants of euclidian routing problems for [2] as well as other references cited in Sec 2). In contrast, our model solves 16 problems spanning (euclidian and non-euclidian) routing, scheduling, packing and graph problems. These problems have diverse structures, compared to the euclidian routing problems, which led previous works to consider different architectures (like [3, 4]). It is in this sense that we claim that, to our knowledge, our approach is the first to tackle such a variety of tasks and structures with a single (backbone) model.
>
> [3] Kwon et al. Matrix Encoding Networks for Neural Combinatorial Optimization, NeurIPS 2021
>
> [4] Pirnay et al. Self-Improvement for Neural Combinatorial Optimization: Sample Without Replacement, but Improvement, Transactions on Machine Learning Research (06/2024)
>
> > The running efficiency is not competent compared with other methods, as shown by Table 1 and 2.
>
> Our method, like all heuristics, provides a trade-off between solution quality and running time. It outperforms most specialized NCO baselines in terms of quality, at the cost of longer running time. In effect, our model is more costly because it is more general, e.g. for CVRP our model takes as input any distance matrix while all the baselines can only take as input the node coordinates. Still in Table 1, we can see that our running times are often only a few seconds and at most 15 seconds, and therefore remain comparable with the baselines.
>
> > Significant gap between GOAL and the optimal solutions as shown in Table 1 and 2.
>
> In Table 1, the gaps between the multi-task-GOAL solutions and the specialized oracle solvers are less than 1% for half the problems and at most of 4.13\%, while GOAL is much faster than the oracles in 7 out of 8 cases (knowing that the oracles run on CPUs and GOAL runs on GPUs). Again, this is about the trade-off between solution quality and running time. While other NCO baselines have similar or larger gaps, we believe it is unrealistic to expect a generalist model to perform on-par with specialized solvers. Similarly in Table 2, we can observe a good quality/time trade-off compared to the baselines as well as the oracle solvers.
>
> > About usefulness for solving real-world problems
>
> In addition to the speed and the ability to leverage GPUs, we believe the main advantage of GOAL in real-world settings is that it can be efficiently fine-tuned to new problems, for which obtaining effective specialized solvers, working at the desired scale, may be too costly.
>
> > Lack of explanations of why the method generalizes to instances of a larger scale better than the baselines
>
> The main difference between our model and the baselines reported in Table 2 (except BQ-NCO) is the use of the BQ-MDPs. And it was shown in the BQ-NCO paper that BQ-MDPs lead to a better generalization. The provided intuition is that the states along a BQ-MDP trajectory are instances which strictly decrease in size at each step, so the model sees a diversity of sizes during pre-training and adapts more easily to new sizes. While in the baselines other than BQ-NCO, the instance component of the state remains constant along a trajectory, hence the model sees less diverse sizes during training.
>
> > Why MatNet cannot be run on ATSP500 or larger, but GOAL can?
>
> MatNet code is publicly available and we observe that it crashes on ATSP500 and larger. Analyzing their implementation, we found that MatNet cannot solve problems larger than its embedding size (256) due to the addition of random identifiers directly into the embedding space. The precise reasons for this design choice would be best clarified by MatNet developers.
>
> > Why GOAL outperforms MatNet, which parts of GOAL contribute to the improvement?
>
> Essentially, MatNet mixes the edge information directly with the vanilla attention score which is the result of the scalar product of two node embeddings. Thus, there is no direct interaction between the edge embeddings and the factors of that product (the node embeddings). Our mixed attention, on the other hand, precisely allows that interaction, since it operates on the factors of the scalar product, as captured by Eq (1b) vs (1a). This may explain the better performance of our mixed attention, as it allows more direct information flow between edges and nodes.

---

> > ### Author Response · Authors · 2024-11-25
> > **Rebuttal to updated review**
> >
> > Following the recent update of the review, we address here the additional concerns:
> >
> > > The neural network of mixed attention has a fatal flaw... When there are no explicit node features, x is initialized as a one-hot vector.
> >
> > No, we do not use one-hot vectors when there are no explicit node features, but random values. As written in L231 of the paper: "we always assume that at least one node feature exists, which distinguishes the nodes by a randomly assigned value between 0 and 1, acting as node identifier, so we always have Ft≥1.". And we fail to see how such a fatal flaw could have been undetected in our numerous experiments.
> >
> > > That is to say, in this case, the scale of the problem N that the model can handle cannot exceed $F_t$, otherwise there would be a reuse of one-hot vectors.
> >
> > Again we disagree, there is no such issue with scaling.
> >
> > > It appears that the model's generalization is severely compromised when $N > F_t$, and the authors have not analyzed this situation.
> >
> > We have shown that our model generalizes to problem sizes N=100, 500, 1000 nodes, while our $F_t$ does not exceed 8. This includes ATSP where there is no explicit node features.
> >
> > > lack of comprehensive analysis over the method's multi-problem solving ability and how the model balances the weights during the learning process for different problems simultaneously to make the model performs best on average.
> >
> > We have added the section "3.6 Training Procedure" to detail our training process. By simply alternating tasks at train time, our model reaches a strong performance, although we are aware of more refined methods that could be investigated in future work. Our ultimate goal is not so much the average performance on the training tasks but rather the ability of the pretrained model to adapt to new tasks.

---

> ### Comment · Reviewer_vQUg · 2024-11-29
> **Thanks for your response**
>
> Authors' rebuttal well addressed part of my concerns on the experiments as well as my previous misunderstandings on the techniques. So I decide to raise the score to 5. However, some of my core concerns have not been adequately addressed, and it seems unlikely that these issues will be resolved satisfactorily in the near term. Consequently, I continue to hold a negative view of this article.
>
> Undoubtedly, this paper presents promising experimental results. My primary concerns, however, revolve around two key issues:
>
> - Theoretical Supports. For instance, there is a lack of compelling theory or analysis to motivate the random assignment of values between 0 and 1 as node features. Furthermore, the capabilities and limitations of such methods is still unknown, including the theoretical boundaries of how well the current framework can generalize to problems of different scales and which types of problems it may not work for.
>
> - Practical Applicapability. Although this work holds the potential to be used as a pre-trained model to tackle various problems in the future, its current version is still quite preliminary.

---

> > ### Author Response · Authors · 2024-12-02
> > **Answer to follow-up concerns**
> >
> > We thank the reviewer for acknowledging our response and summarizing their remaining core concerns. We would like to provide the following clarifications:
> >
> > >  lack of theory or analysis to motivate the random assignment of values between 0 and 1 as node features
> >
> > Indeed we have not motivated this choice in the paper; we will be happy to add a summary of the following arguments in the final version of the manuscript. The idea of adding random node identifiers to distinguish nodes, especially in the absence of other node features, is well known for improving the performance of graph neural networks. For example, [1, 2] prove that adding such random features improve the expressiveness of GNNs. We believe that at least the intuition extends to transformer models. In our early experiments, we have observed that indeed adding these random node identifiers slightly improves the performance. This idea was also previously explored in the MatNet paper, as an alternative to their one-hot node representation which limit the generalization, as noted previously by the reviewer.
> >
> >
> > > lack of theoretical boundaries of how well the current framework can generalize to problems of different scales
> >
> > We demonstrate the generalizability of our model through extensive experiments (results in Table 2). The primary focus of our paper is to explore how a single model can tackle a variety of CO problems, rather than specifically addressing generalization to larger problem sizes. Besides, many prior works focused on scaling to larger instances (e.g., [3, 4, 5, 6]) have not provided such theoretical bounds on their generalization ability.
> >
> > That being said, while we have shown that our model generalizes to instances of approximately 1000 nodes (i.e. ten times bigger than seen in training), it probably won't generalize to much bigger instances. The reason is the quadratic complexity of the underlying transformer architecture. This is a common limitation for all similar transformer-based models, even the ones which specifically target a strong generalization, such that [7, 8]. We mention in Appendix E a few directions to alleviate this limitation, which could be explored in future work.
> >
> >
> > [1] Abboud et al, The Surprising Power of Graph Neural Networks with Random Node Initialization. IJCAI 2021
> >
> > [2] Sato et al, Random Features Strengthen Graph Neural Networks, SDM 2021
> >
> > [3] Son et al, Meta-SAGE: Scale Meta-Learning Scheduled Adaptation with Guided Exploration for Mitigating Scale Shift on Combinatorial Optimization. ICML 2023
> >
> > [4] Pan et al, H-TSP: Hierarchically Solving the Large-Scale Travelling Salesman Problem. AAAI 2023
> >
> > [5] Giu et al, DIMES: A Differentiable Meta Solver for Combinatorial Optimization Problems, Neurips 2022
> >
> > [6] Li et al, Learning to delegate for large-scale vehicle routing. Neurips 2021
> >
> > [7] Drakulic et al, BQ-NCO: Bisimulation Quotienting for Efficient Neural Combinatorial Optimization. Neurips 2023
> >
> > [8] Luo et al, Neural Combinatorial Optimization with Heavy Decoder: Toward Large Scale Generalization. Neurips 2023

---

### Official Review · Reviewer_vjMs · 2024-10-31

**Soundness:** 2
**Presentation:** 3
**Contribution:** 3
**Rating:** 6
**Confidence:** 4

**Summary:**

The paper studies the problem of training a joint multi-task model on multiple combinatorial optimization tasks (COP).
To this end, the GOAL architecture is proposed. This (graph) transformer architecture learns a shared transformer backbone that is shared across COPs. It further maintains COP-specific adaptors that are applied before and after the backbone.
The experiments show that the model trained on multiple tasks is competitive and yields slightly worse than single-task models of the same architecture. It is further shown that fine-tuning a pretrained multi-task model to new tasks yields significantly faster convergence than training a single-task model from scratch.

**Strengths:**

* The problem of multi-task pretraining on multiple CO problems is novel and interesting.
* The main experiments are fairly comprehensive and consider a wide range of problems.
* The reported results are promising and suggest that pretraining can significantly improve convergence speed when fine-tuning for a new problem.

**Weaknesses:**

I am skeptical about the architecture design for multi-type problems, which is why I currently rate this work as a borderline reject. If I understand correctly, two changes are made when working on a multi-type problem:
1. Multiple adaptors are learned, one for each node type (and edge type), respectively.
2. The same backbone model is applied separately to each type and each node type now applies the attention module twice: once to self-attend to other nodes of the same type and a second time to cross-attend to nodes of other types.

The first modification seems useful to map the features of each type to different positions in the latent space. However, it is not clear to me why the second change is helpful. Is this superior to applying the single-type backbone to embeddings produced by heterogeneous adaptors? If so, why? The ablation study seems to combine both changes at once, so the individual contribution of either modification is not demonstrated.

I also think the illustration of the multi-type mechanism in Figure 1 is a bit misleading as it suggests that the output of the self-attention is passed as input to the cross-attention. However, the source code applies both operations in parallel and sums the results.

**Questions:**

* How exactly does the proposed multi-type architecture improve performance? (see Weakness section)
* Considering that the multi-task architecture performs slightly worse than the single-task models, have the authors considered using some multi-task learning techniques (GradNorm, etc...) to potentially close this gap?
* For the fine-tuning experiments, how different would the performance be if one freezes the backbone and only trains the adaptors?

---

> ### Author Response · Authors · 2024-11-25
> **Rebuttal**
>
> We are very grateful for the reviewer's precise and constructive feedback. We answer each concern as follows:
>
> > The ablation study seems to combine both changes at once, so the individual contribution of either modification is not demonstrated.
>
> The first modification is actually unavoidable. Let's say we have $M$ machines with $m$ features and $J$ jobs with $j$ features. Consider the two options:
>
> (i) we concatenate the features of the different types, using zero padding when necessary, then go through the input adapter of size $(m{+}j){\times}\ell$, then the codebook ($\ell{\times}D$) and ending up with an embedding vector of size $(M{+}J) {\times} D$, that we can split into the machines embedding $M{\times}D$ and the jobs embedding $J{\times}D$.
>
> (ii) use type-specific adapters of size $m{\times}\ell$ and $j{\times}\ell$, then apply the codebook and end up again with the machines embedding $M {\times} D$ and the jobs embedding $J{\times}D$.
>
> Because of the linearity of the adapters and the codebook, options (i) and (ii) are strictly equivalent. That is why our ablation only evaluates the advantage of the multi-type versus single-type architecture (the second change below). We added this clarification in the ablation section.
>
> > Why the second change is helpful. Is this superior to applying the single-type backbone to embeddings produced by heterogeneous adaptors?
>
> The ablation study focuses on testing the impact of the second change; i.e. comparing using the single-type model on the $(M{+}J){\times}D$ embeddings versus the multi-type model on the $M{\times}D$ and $J{\times}D$ embeddings. It shows that the later leads to a superior performance. One reason may be that in the multi-type attention blocks, the softmax operation which defines the attention coefficient scopes only over nodes of the same type. This means that the transformation can spread attention weight independently on jobs and on machines, instead of having to spread it on both jointly as in the single-type model, leading to a rather counter-intuitive competition between types for attention.  Furthermore, the multi-type model allows to define smaller blocks, that can be adapted to the "structure" of the type: for example for the UMSP, as shown in Figure 7 (a), the type-specific self-attention blocks are vanilla attention modules, leveraging the fact that there are no edges between the jobs (resp. machines). Finally, the multi-type model offers more flexibility for  fine-tuning: indeed, although it is crucial that the different blocks share the same parameters at training (to remain task agnostic), this constraint can be relaxed at fine-tuning time to better fit the specific task at hand.
>
> > How exactly does the proposed multi-type architecture improve performance?
>
> See above.
>
>
> > Illustration of the multi-type mechanism in Figure 1 a bit misleading...
>
> Thank you very much for pointing this out, the figure actually corresponds to a previous implementation, which we found to be slightly slower. We have updated it in the manuscript and provide additional illustrations in Figure 7 (Appendix D). In any case, because we stack several layers, we observed that what is important is not so much how the attention blocks are connected in each layer, but the fact that they all share the same parameters (within a layer), so that the model remains generic.
>
>
> > Multi-task learning techniques (GradNorm, etc...)
>
> We experimented with advanced multitask training approaches, including meta-learning (a Reptile-inspired) methods and gradient aggregation techniques similar to GradNorm. However, these approaches did not outperform our standard multitask training. The primary challenge lies in the computational expense of training the GOAL model, which makes parameter tuning for these pipelines particularly costly. We plan to investigate these methods further, as this represents an important direction for future research.
>
> > Backbone freeze when finetuning
>
> We have updated Fig. 5 with the performance when the backbone is frozen. It concerns the supervised fine-tuning mode, but the results are essentially the same in the unsupervised case (Fig. 2). As one may expect, freezing the backbone yields a degraded performance. Freezing the backbone (or limiting its finetuning to internal adapters as in LoRa) is common practice and makes sense when dealing with very large models such as LLMs. But it is to be noted that our GOAL backbone is relatively small, so the computational impact of fine-tuning it in full is almost negligible. That is also why we are not using adapters (e.g. LoRa) inside the backbone.

---

> > ### Comment · Reviewer_vjMs · 2024-11-26
> >
> > I appreciate the authors' clarifications and adjustments to the manuscript.
> >
> > I have one follow-up question that remains unclear to me at this point: How exactly are problems with more than two types handled? Say there are n types. Would each vertex apply the attention module twice (once to self-attend to its own type and once to cross-attend to all other types at once) or n times (once for each individual type)? If it is the latter, would the results of all attention operations simply be summed? How GOAL works in this setting or whether it is restricted to at most two types is not fully clear in the current version of the manuscript. If this was improved, then I would increase my rating of this paper.

---

> > > ### Author Response · Authors · 2024-11-26
> > > **Response to follow-up question**
> > >
> > > Thank you very much for acknowledging our rebuttal and we appreciate the follow-up question.
> > >
> > > > Multi-type GOAL for $n$ types:
> > >
> > > The natural approach with $n$ types is to define $n^2$ mixed-attention blocks: each type attends to itself and every other type. In practice, it is unlikely that all the type combinations are relevant to the task, so we may have less cross-attention blocks. In general, the $n$ outputs for each type would be summed (or averaged) before going through the normalization and feed-forward layers, resulting in $n$ output embeddings (one per type). We have added this clarification in the updated manuscript (see highlighted text in Section 3.5).

---

> > > > ### Comment · Reviewer_vjMs · 2024-12-02
> > > >
> > > > I thank the authors for the additional clarifications and appreciate the paper improvements. I have raised my rating score to 6.

---

### Official Review · Reviewer_QzXG · 2024-11-03

**Soundness:** 4
**Presentation:** 3
**Contribution:** 4
**Rating:** 8
**Confidence:** 4

**Summary:**

This paper presents a new transformer-like architecture, named GOAL, for general types of combinatorial optimization training. GOAL is an auto-regressive model with a shared architecture across different problem types with specialized input and output layers for each problem. This architecture learns a generalizable rule that solves different COs and can generalize to new problems by fine-tuning. Experiment result shows that GOAL could be comparative or outperform problem-specific state-of-the-arts (greedy version).

**Strengths:**

* A generalized combinatorial optimization solver is favored by the research community.
* The proposed GOAL transformer architecture seems interesting and promising, especially given the fact that it outperforms other neural networks when trained for a specific problem.
* The generalized training and fine-tuning result seems sound and promising. The greedy version of GOAL is comparative to other greddy peer methods.

**Weaknesses:**

Some important details are missing in this paper:
* What are the implementation details of the "codebook"? Please specify
* Definitions of  BQ-MDP and tail-recursive are needed in the main text to make this paper self-contained.

Misc:
* The first paragraph is too long
* There are multiple misuses of \citep and \citet in this paper. For example, In L112, please use \citet for Khalil et al., 2017. In L116, please use \citep for Kool et al. (2019). Please proofread and fix all the misusages
* If the oracle solver is not an optimal solver, the results should not be claimed as "optimal gap" in Table 1

**Questions:**

The following questions should not be considered as "weaknesses", but I believe are worth discussing:
* What is the parameter size of GOAL and how does it compare to other peer methods?
* How to implement the post-processing step (such as MCTS, 2-opt search) that has been proven to be prominent for neural network CO solvers?
* For problems where there could be different ways of defining nodes and edges, how will different node and edge definitions affect the solver's performance?
* To what extent does the current version of GOAL scale up to larger-sized problems?

---

> ### Author Response · Authors · 2024-11-25
> **Rebuttal**
>
> We thank the reviewer for their insightful and constructive feedback. We address your concerns as follows:
>
> > Implementation details of the "codebook"
>
> The codebook is a (low-rank) matrix of size $\ell{\times}D$), shared by all tasks, and mapping the output of the input adapter (of dimension $N{\times}\ell$) to the input of the backbone ($N{\times}D$). We added more details at the end of Sec 3.4. The term codebook is usually reserved for a discrete set of representations (here $\ell$ embedding vectors of size $D$), and we abuse it slightly by allowing each input adapter to represent its task features by linear combinations of the codebook vectors rather than single ones.
>
> > Definitions of BQ-MDP and tail-recursive are needed in the main text
>
> We added a couple of sentences at the end of Sec 3.1 to highlight the main difference between the BQ-MDPs and those more commonly used in other NCO frameworks, as well as more information on tail recursivity.
>
> > Parameter size of GOAL and how does it compare to other peer methods
>
> We added the following precision in Appendix A. The size of our model is in the same range as peer models. The GOAL backbone has 2.1M parameters and adapters are very light, with only a few thousand parameters per task. For comparison, RouteFinder+POMO has 1.3M, MVMoE has 3.7M (the same model is used in RouteFinder+MVMoE), and RouteFinder+Transformer_Encoder 1.7M.
>
> > How to implement the post-processing step (such as MCTS, 2-opt search)
>
> Post-hoc improvements can be applied exactly as in similar constructive NCO works (e.g. beam search in AM [2] and BQ-NCO [3], MCTS in [4]). Alternatively, after constructing the greedy solution, one can apply 2-opt or any other improvement heuristic.
>
> [2] Kool et al, Attention, Learn to Solve Routing Problems! ICLR 2019
>
> [3] Drakulic et al, BQ-NCO: Bisimulation Quotienting for Efficient Neural Combinatorial Optimization, Neurips 2023
>
> [4] Xing et al, A Graph Neural Network Assisted Monte Carlo Tree Search Approach to Traveling Salesman Problem, IEEE Access 2020
>
> > How will different node and edge definitions affect the solver's performance?
>
> Some problems can indeed be represented equivalently with various node and/or edge features. For example, in the Euclidean routing problems, location information can be represented either as node features (using node coordinates) or as edge features (using a distance matrix). We chose the latter approach for all routing problems because it is more general. When GOAL is used for a single routing task like TSP, performance is slightly superior when using distance (injected with GOAL's mixed attention) rather than coordinates (using vanilla attention). Table 1 reports a .3% opt gap on TSP100 for the former while the BQ-NCO paper reports .35% for the latter (single-task GOAL boils down to BQ-NCO when coordinates are used since edge information is void in that case).
>
> > Scaling to larger problems
>
> Generalization results on larger instances, with up to 1000 nodes, are available in Table 2 in the Appendix. Originally we provided the generalization results for 3 problems, we added 3 more in the updated manuscript. Note that the attention bottleneck can be mitigated by considering task-specific heuristics to prefilter the input graph (see Sec 3.4), allowing generalization to even larger sizes on some tasks.

---

### Official Review · Reviewer_5MgC · 2024-11-09

**Soundness:** 3
**Presentation:** 3
**Contribution:** 3
**Rating:** 6
**Confidence:** 4

**Summary:**

This paper introduces a generalist model designed to address various combinatorial optimization problems. Unlike traditional machine learning approaches, which require a specialized and separately trained model for each problem, this method utilizes a shared backbone network with lightweight, problem-specific adapters for input and output processing. The backbone incorporates mixed-attention blocks that accommodate different combinations of node, edge, and instance-level features, while a multi-type Transformer architecture handles heterogeneous node and edge types. Experiments show that this method performs nearly as well as specialized models in a multi-task setting across diverse problems and demonstrates strong transfer learning capabilities, adapting effectively to new problems through fine-tuning.

**Strengths:**

（1）The paper is novel , as it designs a multi-task learning approach to solve various combinatorial optimization problems through an end-to-end model. The authors developed a mixed-attention block to effectively achieve this objective.

（2）The paper is well-organized, concisely written, and has good readability.

（3）This paper demonstrates substantial work, conducting experiments on various combinatorial optimization problems and showcasing the effectiveness of the proposed method in terms of solution quality and speed.

**Weaknesses:**

（1）The description of dimension transformations and the learning process of the model is not very illustrative. It is recommended to add figure and text to enhance the explanation.

（2）In Table 1, only one problem size is tested, and it is relatively small. It is recommended to include experiments with larger problem sizes.

（3）The paper lacks a theoretical analysis of the method's effectiveness, and it is recommended to include this section.

（4）There are few effective baselines in Table 1. For ATSP, CVRPTW, OP, KP, MVC, and JSSP, there is only one or two baselines, and UMSP lacks an NCO baseline, which weakens the convincing power. It is recommended to add more baselines.

**Questions:**

（1）Please address the issues raised in the weaknesses section.

（2）The paper assumes that solving strategies for various combinatorial optimization problems share common knowledge. How can it be proven that such knowledge exists, and how can the overlap of this knowledge across different combinatorial optimization problems be demonstrated?

---

> ### Author Response · Authors · 2024-11-25
> **Rebuttal**
>
> We thank the reviewer for the insightful feedback and address their concerns and questions as follows:
>
> > Description of dimension transformations and the learning process
>
> To complement the dimensions that are explicit in Fig 1, we added the following at the end of Sec 3.4:
>
> "... node (resp. edge) features are first mapped into a low dimension $\ell{\ll}D$ (resp. $\bar{\ell}{\ll}\bar{D}$), through a task-specific linear projection (the input adapter proper), forcing the representations to share dimensions across tasks. Then this "small" representation is plunged into a "large" $D$- (resp. $\bar{D}$)-dimensional embedding, through a common linear projection ..."
>
> Besides, we added a new section "3.6 Training Procedure" to detail the learning process.
>
> > Experiments with larger problem sizes
>
> In Table 1, we provide the results on instances with 100 nodes, coming from the same distribution as the training instances. Generalization results on larger instances, with up to 1000 nodes, are available in Table 2, presented in the Appendix for space reasons. Originally we provided the generalization results for 3 problems, we added 3 more in the updated manuscript.
>
> > Lack of theoretical analysis
>
> We indeed do not provide a theoretical analysis but demonstrate the effectiveness of our approach through extensive experiments. In addition to the performance on the training tasks (Tables 1 and 2) and fine-tuning on new tasks (Fig 2), we empirically demonstrate the synergistic effect of our proposed multi-task pre-training for effective fine-tuning on new tasks in Fig 4. We want to point out that similar works that aimed at foundation models in other domains, such as GATO [1], do not provide a theoretical analysis either.
>
> [1] Reed et al., A Generalist Agent, Transactions on Machine Learning Research (11/2022)
>
> > Lack of (NCO) baselines for certain problems in Table 1
>
> We included extra baselines for CVRPTW and JSSP in Table 1. Regarding other problems, we are not aware of more NCO baselines, and in particular we do not know any for the USMP.
>
> >  Existence of common knowledge between different CO problems
>
> We implicitly conjecture but do not prove the existence of such a common knowledge. Intuitively, some CO problems share common underlying concepts/skills like shortest paths for routing problems, and therefore we may think pre-training on a set of problems may enable the model to learn such skills which will then be helpful to effectively adapt to a new problem. The experimental results tend to confirm this intuition. Indeed, if there was no common knowledge acquired during the training, finetuning the pre-trained model on a new task would be equivalent to learning it from scratch, which is not what we observe (Fig 2).

---

### Meta-Review · Area_Chair_yHj3 · 2024-12-17

**Metareview:**

This paper presents GOAL, a generalist model for combinatorial optimization. It employs a shared backbone with problem-specific adapters and a multi-type transformer architecture. The model is trained on multiple problems and shows effectiveness in solving various COPs and strong transfer learning capabilities through fine-tuning.

Strengths include its innovative generalist approach, clear organization, and extensive experimentation. However, it has weaknesses such as insufficient theoretical analysis, unclear explanations in some aspects like dimension transformations, and limited baselines in experiments. The model's running efficiency and the gap to optimal solutions also need attention. Overall, the paper's novel concept and the authors' efforts to address concerns during the rebuttal warrant its acceptance, as it has the potential to contribute valuable insights and advancements to the field of combinatorial optimization.

**Additional Comments On Reviewer Discussion:**

During the rebuttal period, reviewers raised several points. These included issues like clarifying model details (e.g., dimension transformations, codebook implementation), expanding experiments (e.g., with larger problem sizes, more baselines), justifying architecture design choices (e.g., for multi-type problems), improving theoretical analysis, and addressing concerns about performance compared to other methods. The authors addressed these by adding text to clarify model workings, conducting more experiments and reporting results, providing justifications for design decisions, discussing the reasoning behind performance differences, and outlining future research directions. In weighing these points, the authors' responsiveness and the value of the proposed generalist model despite its areas for improvement led to the decision to accept the paper.

---

### Decision · Program_Chairs · 2025-01-22

Accept (Poster)